# Dopaminergic neurons write and update memories with cell-type-specific rules

Yoshinori Aso*, Gerald M Rubin*

Janelia Research Campus, Howard Hughes Medical Institute, Ashburn, United States

**Abstract** Associative learning is thought to involve parallel and distributed mechanisms of memory formation and storage. In Drosophila, the mushroom body (MB) is the major site of associative odor memory formation. Previously we described the anatomy of the adult MB and defined 20 types of dopaminergic neurons (DANs) that each innervate distinct MB compartments (*Aso et al., 2014a*, *2014b*). Here we compare the properties of memories formed by optogenetic activation of individual DAN cell types. We found extensive differences in training requirements for memory formation, decay dynamics, storage capacity and flexibility to learn new associations. Even a single DAN cell type can either write or reduce an aversive memory, or write an appetitive memory, depending on when it is activated relative to odor delivery. Our results show that different learning rules are executed in seemingly parallel memory systems, providing multiple distinct circuit-based strategies to predict future events from past experiences.

*For correspondence: asoy@
janelia.hhmi.org (YA); rubing@
janelia.hhmi.org (GMR)

**Competing interests:** The authors declare that no competing interests exist.

## Introduction

Animals use memories of past events to predict the future. In some cases, an animal is best served by making a prediction based solely on their most recent experience. In others, a series of experiences is integrated to make a probabilistic prediction, discounting an event experienced only once. How are such different strategies implemented in the brain? One idea is that individual components of a memory—often called engrams—are simultaneously stored in distinct sub-circuits whose outputs can then be combined upon recall to affect behavior. These sub-circuits would vary in their rules for writing, updating and retaining these engrams, having differences in synaptic plasticity and circuit properties (*Hikosaka et al., 2014*). Experiments aimed at uncovering the mechanisms by which different forms of memory are established and maintained, and then coherently coordinated to drive behavior, are facilitated by using a model system in which the relevant cells and circuits can be identified and manipulated either individually or in specific combinations. In this report, we describe experiments performed in such a model system, the olfactory circuitry of *Drosophila melanogaster*.

Neuronal circuits for learning associations often share a common architecture: a large array of anatomically similar neurons that represent the sensory environment converge onto a much smaller number of output neurons (*Luo, 2015*, *Dean et al., 2010*) (*Figure 1A*). Punishment or reward activates modulatory neurons that in turn cause changes in the synaptic weight matrix between the neurons representing the sensory cues and the output neurons, resulting in memory formation. The mushroom body (MB) shares this architecture and is the major site of associative learning in Drosophila (*Heisenberg et al., 1985*; *de Belle and Heisenberg, 1994*; *Dubnau et al., 2001*; *McGuire et al., 2003*; *Heisenberg, 2003*). Odor identity is represented by a pattern of sparse activity in the ~2000 Kenyon cells (KCs) (*Laurent and Naraghi, 1994*; *Perez-Orive et al., 2002*), whose parallel axons form the lobes of the MB. The dendrites of MB output neurons (MBONs) and terminals of dopaminergic input neurons (DANs) tile the KC axons; the extent of the arbors of individual

MBONs and DANs overlap precisely, defining 15 distinct compartmental units that tile the MB lobes of adult Drosophila (*Tanaka et al., 2008*; *Aso et al., 2014a*).

While these compartmental units share a similar general structure, there are important anatomical and functional differences between them. Each compartment contains only one of the three major classes of KCs: γ, α′/β′ and α/β. Anatomical data suggest that each MBON samples from ~90 to ~2000 KCs in one or two compartments, while each KC forms *en passant* synapses on 5–6 types of MBONs along its axon in the lobes (*Aso et al., 2009*, *2014a*). During associative learning, each DAN is thought to modulate KC-MBON synapses only in its target compartment(s) (*Hige et al., 2015*; *Cohn et al., 2015*). Thus, the information encoded by KC activity might contribute to multiple distinct engrams by compartmental specific modulation of KC-MBON *en passant* synapses. A large body of previous work has established that the KCs, MBONs and DANs innervating individual compartments are differentially involved in forming memories with different valence—that is, appetitive and aversive memories—and with different stabilities (*Schwaerzel et al., 2003*; *Zars et al., 2000*; *Isabel et al., 2004*; *Blum et al., 2009*; *Krashes et al., 2009*; *Claridge-Chang et al., 2009*; *Aso et al., 2010*; *Sejourne et al., 2011*; *Trannoy et al., 2011*; *Liu et al., 2012*; *Burke et al., 2012*; *Aso et al., 2012*; *Placais et al., 2013*; *Pai et al., 2013*, *Aso et al., 2014b*; *Lin et al., 2014*; *Bouzaiane et al., 2015*; *Ichinose et al., 2015*; *Yamagata et al., 2015*; *Owald et al., 2015*; *Huetteroth et al., 2015*). However little is known about the rules for writing and updating memory in each compartment. By using the anatomical map of the MB and cell type specific drivers we reported previously (*Aso et al., 2014a*, *2014b*) in conjunction with newly developed behavioral assays, we have been able to establish that different compartments can employ vastly different learning rules.

## Results and discussion

### Experimental design

Punitive or rewarding stimuli such as electric shock, heat, cold, bitter taste and sugar generally activate a complex pattern of DANs (*Riemensperger et al., 2005*; *Mao and Davis, 2009*; ; *Liu et al., 2012*; *Tomchik, 2013*; *Galili et al., 2014*; *Das et al., 2014*; *Kirkhart and Scott, 2015*). Believing that a reduction in complexity will be essential to understand the roles played by DAN inputs to different MB compartments, we utilized intersectional split-GAL4 drivers to express CsChrimson, a red-shifted channelrhodopsin, in specific cell types (see Materials and methods). While activation of CsChrimson in these driver lines provides a stimulus that is unlikely to occur naturally, it allowed us to separately examine the memory components induced by individual DAN cell types.

We developed an olfactory arena that allows fine temporal control of both odor delivery and DAN activation using optogenetics (*Figure 1B*, *Video 1* and *Figure 1—figure supplement 1*; detailed construction documents are included as a *Supplementary file 1*). In this arena, freely moving flies can be repeatedly trained and tested without the manual handling or temperature changes required in previous assays, thereby minimizing variability that might obscure subtle behavioral effects. These methods allowed us to systematically examine the properties of memories induced in different MB compartments, including: (1) the temporal pairing requirements of odor presentation and DAN activation; (2) the amount of training required for memory formation; (3) retention time; (4) weakening of the conditioned response induced by either DAN activation in the absence of odor presentation or by odor presentation in the absence of DAN activation; (5) the ability to learn new associations; and (6) the capacity to store multiple memories.

### Assessing the learning rules in one compartment

Dopamine signaling to KCs has been implicated in both learning and forgetting (*Schwaerzel et al., 2003*; *Schroll et al., 2006*, *Kim et al., 2007*; *Tomchik and Davis, 2009*; *Gervasi et al., 2010*; *Qin et al., 2012*; *Placais et al., 2012*; *Berry et al., 2012*; *Boto et al., 2014*; *Shuai et al., 2015*). However, it has not been determined if a single DAN cell type can drive both processes within the same compartment. Pairing one of two odors with activation of PPL1-γ1pedc, results in robust aversive memory to the paired odor (*Aso et al., 2010*; *Figure 1C–E* and *Video 2*). That memory is fully retained after 10 min (blue line in *Figure 1D*) but has largely decayed by 24 hr (Figure 3B). Presentation of the odors alone a few minutes after training resulted in a modest reduction in the

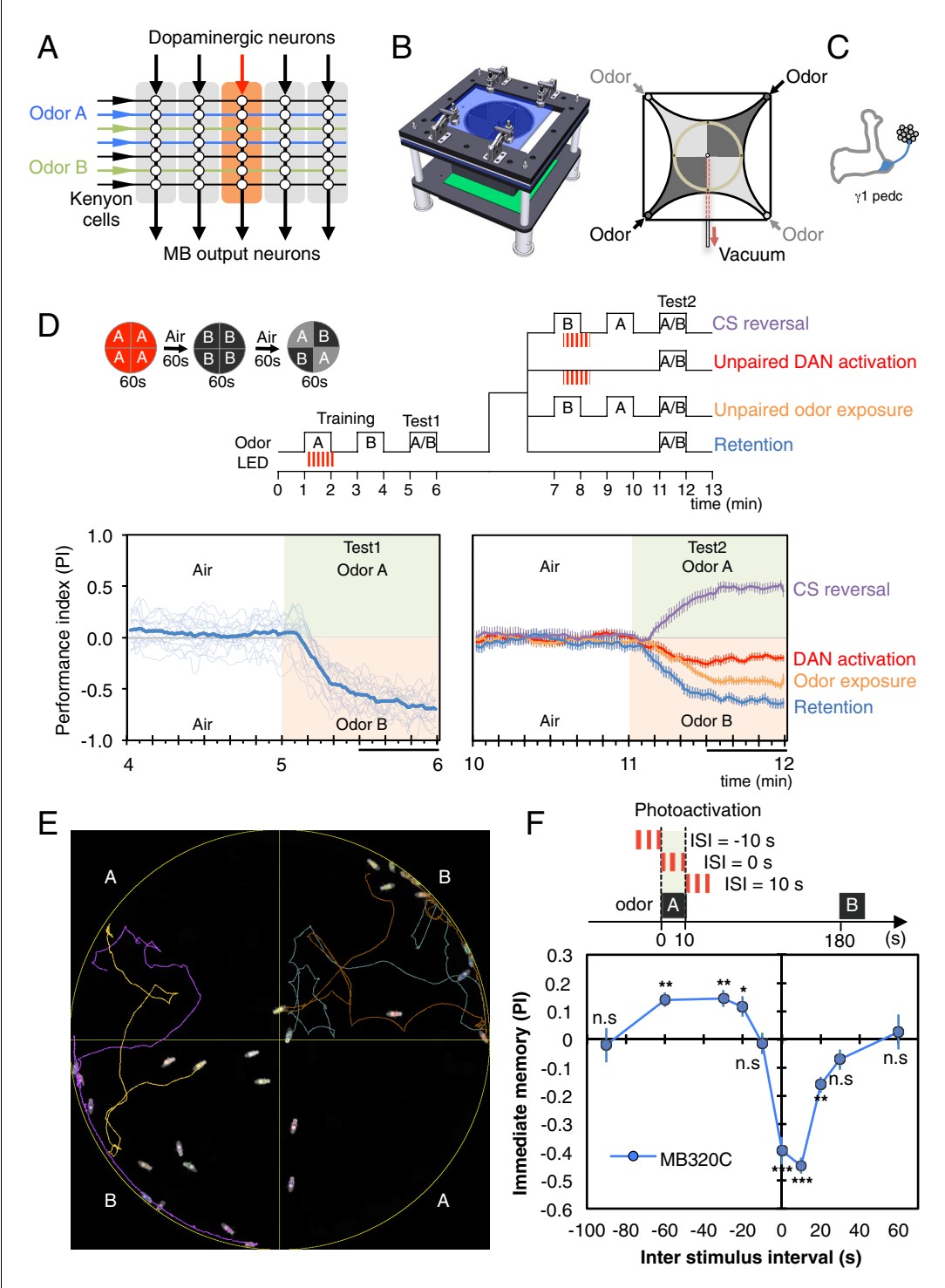

**Figure 1.** Learning rules in one MB compartment. (**A**) Conceptual diagram of memory circuit organization. The parallel axonal fibers of the Kenyon cells represent odor stimuli, modulatory dopaminergic inputs induce plasticity at KC to MBON synapses in distinct compartments (represented by the shaded rectangles) along the length of these axons and MB output neurons from the compartments read out memory. See *Aso et al. (2014a)* for more details. (**B**) Left: Design of the optogenetic olfactory arena. See *Figure 1—figure supplement 1* and *Supplementary file 1* for details. Right: Diagram illustrating odor paths in the arena. Flies are confined in the circular arena at the center (3-mm high and 10-cm diameter). *Video 1* illustrates the pattern of airflow. (**C**) Diagram of the expression pattern of the split-GAL4 line MB320C driving CsChrimson-mVenus (blue) in PPL1-γ1pedc; see *Aso et al. (2014a)* and www.janelia.org/split-gal4 for primary image data for this and other split-GAL4 lines. (**D**) Top: Training protocols. For odor delivery, valves were open for 60 s. For training, thirty 1 s pulses of red light (627 nm peak and 34.9 μW/mm² at the position of the flies) were applied over 60 s starting

*Figure 1 continued on next page*

*Figure 1 continued*

5 s after valve opening. Experiments were done reciprocally: In one group of flies, odor A and B were 3-octanol and 4-methylcyclohexanol, respectively, while in a second group of flies, the odors were reversed. All flies went through the same initial training and test protocol, and then the flies, without removal from the arena, went through one of the four diagramed training and test protocols. Bottom: Time course of the performance index (PI) during first test period (from 4–6 min of the experiment; left) and second test period (from 10–12 min of the experiment; right). The PI is defined as [(number of flies in the odor A quadrants) - (number of flies in odor B quadrants)]/(total number of flies). The average PI of reciprocal experiments is shown. The overall PI, which is reported in *Figure 1F* and *Figures 2* and *3* was calculated by averaging the PIs from the final 30 s of each test period (indicated by the black horizontal line on the time axis). In the left panel, thin lines show individual reciprocal experiments and the thick line the mean of all experiments. In the right panel the mean with error bars representing the SEM are shown. Control genotypes did not show any significant memory in the same training protocols: (1) no driver control, *pBDPGAL4* in *attP2/20xUAS-CsChrimson* in *attP18* (PI = 0.07, SEM = 0.037, N = 14); and (2) no effector control, *MB320C/w1118* (PI = −0.01, SEM = 0.031, N = 10). (E) A single frame of *Video 2* showing the position of odor-conditioned flies at the end of the 1-min test period. Lines show trajectory of four flies. *Video 2* shows the behavior of flies in the area for the full 1 min test period. (F) Inter-stimulus-interval (ISI) curve. A single training was done for each experiment. The relative timing of a 10 s delivery of odor A and a 10 s period in which three 1 s light pulse were delivered was varied. The diagram on top illustrates the cases of ISI = −10 s, 0 s and +10 s corresponding to the light pulses starting −10 s, 0 s, or +10 s after the initiation of odor delivery, respectively. The data points and error bars show the mean and SEM for MB320C/CsChrimson-mVenus. N = 10–14. Asterisk indicates significance from 0: *p<0.05; **p<0.01; ***p<0.001; n.s., not significant.

The following figure supplement is available for figure 1:

**Figure supplement 1.** Diagram of the olfactory behavioral apparatus.

conditioned response (orange line in *Figure 1D*). A second activation of the same DAN a few minutes after training in the absence of odor can almost completely abolish the conditioned response (red line in *Figure 1D*). Recent imaging data of the γ4 MBON (*Cohn et al., 2015*) suggest that this reduction most likely results from restoration of the response of the MBON to the odor; that is, erasure of the memory. If, after the first training, contingencies are reversed such that the other odor is presented paired with DAN activation, the first memory is reduced and a memory of the new association is formed (purple line in *Figure 1D*). Taken together, these data indicate that the same DANs can write a new memory or reduce an existing conditioned response, enabling the flexibility to rapidly change the associations formed between a conditioned stimulus (CS), the odor, and an unconditioned stimulus (US) represented by dopamine release.

In classical conditioning, both the rate of learning and the valence of the resultant memory depend on the relative timing between the CS and the US (*Christian and Thompson, 2003*; *Gerber et al., 2014*). Our ability to precisely control DAN stimulation and odor presentation enabled us to examine the CS-US timing relationship (*Figure 1F*). We found that when PPL1-γ 1pedc stimulation fell within a 30-s time window following the onset of a 10-s odor presentation, an aversive memory was formed (negative PI, *Figure 1F*). Interestingly, we observed that DAN stimulation that precedes odor presentation by 20 to 60 s induced an appetitive memory (positive PI, *Figure 1F*). These observations are consistent with the notion that it is the predictive aspect of CS-US timing that matters. When the timing is such that the CS predicts a subsequent aversive US, animals learn to avoid the CS. However, animals can also learn that the CS predicts the end of an aversive US, and are consequently attracted to the CS (*Tanimoto et al., 2004*). It has been suggested that this timing dependency could result from the dynamics of the

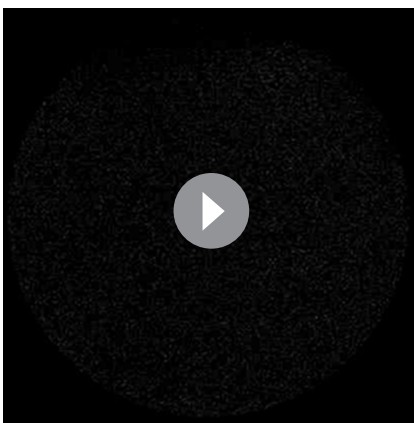

**Video 1.** Separation of airflow in the four quadrants of arena. Ammonium chloride smoke was introduced in two opposing quadrants allowing the borders of airflow in the circular arena to be seen. Valves opened at the beginning of the movie. After a ~2 s delay, smoke reached the peripheral of arena, a further ~3 s was required to fill the arena. At a flow rate of 400 mL/min, replacing the 23.5 mL of air in the arena takes ~3.5 s. Elapsed time after valve opening is shown in the upper right.

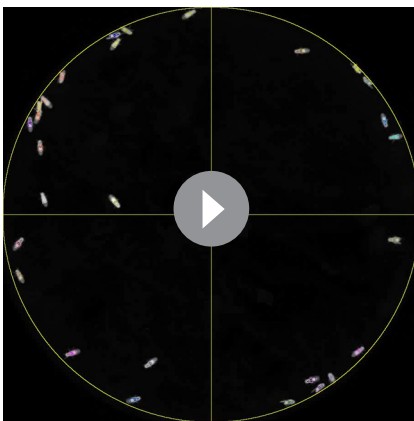

**Video 2.** Conditioned odor avoidance after training with PPL1-γ1pedc. An example of conditioned odor avoidance behaviors after training using optogenetic activation of PPL1-γ1pedc. The movie shows the 1-min test period (corresponding to the 5 to 6 min time range in *Figure 1D* left). Colored lines follow the trajectories of four flies. Notice that flies approaching the borders between compartments readily cross into the B quadrants with the control odor, but avoid crossing into the A quadrants with the trained odor.

biochemical signaling cascade acting downstream of dopamine receptors (*Yarali et al., 2012*).

Pairing activation of PPL1-γ1pedc with odor has been shown to depresses the subsequent spiking rate of the MBON from the γ1pedc compartments in response to the trained odor (*Hige et al., 2015*). In behavioral assays, optogenetic activation of this MBON was shown to attract flies (*Aso et al., 2014b*). Taken together, these observations suggest that DAN activation paired with an odor produces an aversive behavioral response to that odor by decreasing the MBON's attractive output. Thus our data can be most easily explained if this single DAN can bi-directionally alter the strength of KC-MBON synapses depending on the presence and relative timing of odor-driven KC activity; a full testing of this model awaits additional physiological measurements.

## Comparing rules for writing memory across compartments

In order to compare the parameters of learning in different MB compartments, we selected a set of additional split-GAL4 drivers that express at similar and high levels in different DAN cell types (diagrammed in *Figure 2A*; primary imaging data documenting expression patterns can be found at http://www.janelia.org/split-gal4). In addition, the three DAN cell types, which express CsChrimson using the MB320C and MB099C drivers, have been shown to have similar spiking responses to CsChrimson activation (*Hige et al., 2015*). In three of six cases, we chose drivers expressing in a combination of two cell types because we found that activation of only a single DAN cell type did not produce a sufficiently robust memory (*Figure 2—figure supplements 1,2*). We confirmed that the lines used in our optogenetic experiments (*Figure 2A*) showed comparable memory formation when trained with electric shock or sugar reward (*Figure 2—figure supplement 3*). Together, the selected drivers innervate 11 of the 15 MB compartments. Below we describe the results obtained in a number of different learning assays by activating these split-GAL4 drivers.

Longer and repetitive training has been shown to induce stronger and more persistent memory across animal phyla (*Frost et al., 1985*; *Tully et al., 1994*). Consistent with those observations, we found that a 10-s training generally induced memory less effectively than a 60-s training in our immediate memory assay (*Figure 2B*). We also found that the optimal temporal relationship of DAN activation and odor presentation for memory formation was similar, but not identical, for DANs innervating different MB compartments (*Figure 2—figure supplement 4*).

Long-term aversive memory in flies requires repetitive electric shock conditioning with resting intervals, so-called spaced training (*Tully et al., 1994*). We found that two sets of DANs, PPL1-α3 alone (MB630B) or the combination of PPL1-γ2α′1 and PPL1-α′2α2 (MB099C) can induce 1-day and 4-day aversive memory after spaced training (*Figure 2C*), suggesting that the effects of spaced training can be implemented in individual compartments. Making memory formation dependent on repetitive training might be beneficial by allowing an animal to ignore spurious one-time events. Recent work has shown that the γ2α′1 compartments play key roles in both sleep regulation and long-term memory (*Sitaraman et al., 2015*; *Haynes et al., 2015*). The observation that co-activation of PPL1-γ2α′1 and other DANs synergistically prolongs memory retention (*Aso et al., 2012*) raises the possibility that PPL1-γ2α′1 might act broadly to facilitate memory consolidation by promoting sleep after learning.

We found that a particular DAN's ability to induce the formation of immediate, 1-day and 4-day memories is not correlated. For example, immediate memory after a single pairing with activation of

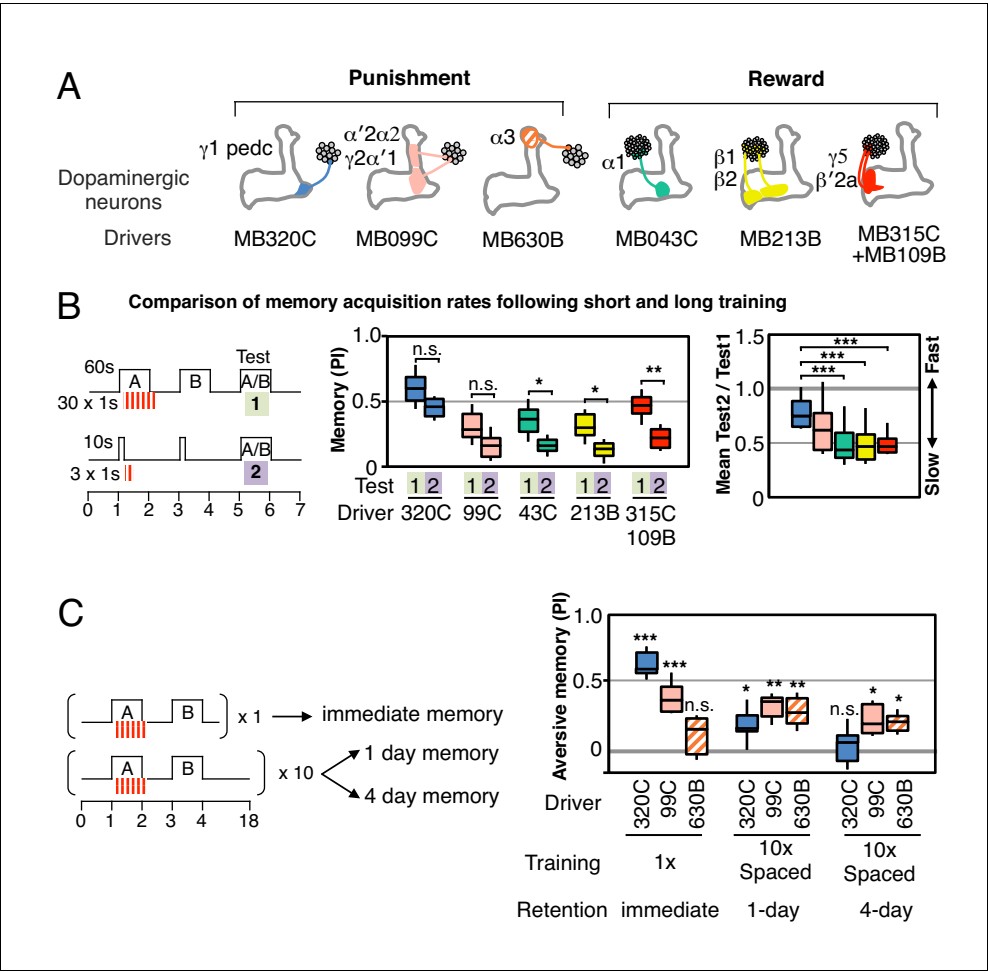

**Figure 2.** Rules for writing memory. (**A**) Diagram of the DANs contained in split-GAL4 driver lines, which have been color-coded to facilitate comparison with the plots shown in the subsequent panels. Expression patterns of these drivers, including full confocal stacks, can be found at www.janelia.org/split-gal4. In all experiments, the drivers were crossed with 20xUAS-CsChrimson-mVenus in attP18. (**B**) Differential effect of training length among DANs. Left: Diagram of the experimental design. Immediate memories formed after paring a 60-s odor presentation with thirty 1-s CsChrimson-activating light pulses (Test 1) were compared with those obtained with a 10-s odor presentation paired with three 1-s light pulses (Test 2). Center: A 60-s training period resulted in significantly better memory performance compared with a 10-s training for MB043C, MB213B and MB315C +MB109B (data from ISI = 0 s for MB099C and ISI =+ 10 s for others drivers were used to provide maximum memory formation; see ISI curves in *Figure 2—figure supplement 4*). We also observed increased learning with 60-s versus 10-s training (PI of 0.72 versus 0.15) when using R58E02-GAL4 (*Liu et al., 2012*), a strong GAL4 driver expressed in ~90 PAM cluster DANs that includes all of the ~50 DANs that have expression in MB043C, MB213B and MB315C+MB109B. To facilitate comparison of PI magnitudes, the sign of the PI in this and subsequent panels was reversed for DANs that induced aversive memory (MB320C, MB099C and MB630B). The bottom and top of each box represents the first and third quartile, and the horizontal line dividing the box is the median. The whiskers represent the 10th and 90th percentiles. N = 8–16. Right: Comparison of the effect of training time on memory formation induced by activation of different DANs. Ratios of the mean PI obtained with short training and individual PIs obtained with long training are shown for each driver. Asterisk indicates significance of depicted pairs after comparing all pairs. (**C**) Comparison of learning after single and repetitive training using the three drivers MB320C, MB099C and MB630B. Either a single training with memory test after 1 min (immediate memory; left) or 10 trainings separated by 15 min resting intervals and then memory tests after 1 (middle) or 4 (right) days were used. Significant aversive 1-day memory was seen with all drivers, while 4-day memory was observed only with MB099C and MB630B. MB320C failed to show 4-day memory despite displaying the most robust immediate memory, while MB630B did not induce significant immediate memory. N = 8–12. Asterisk indicates significance of comparison of indicated pairs in B and from 0 in C: *p<0.05; **p<0.01; ***p<0.001.

*Figure 2 continued on next page*

*Figure 2 continued*

The following figure supplements are available for figure 2:

**Figure supplement 1.** Combinatorial roles of DANs in memory formation.

**Figure supplement 2.** Additional drivers that induced weak, but significant, memory.

**Figure supplement 3.** Controls for genetic background.

**Figure supplement 4.** Inter stimulus interval curves.

**Figure supplement 5.** A conceptual model of memory dynamics in parallel memory units.

PPL1-$\alpha$3 (MB630B) was barely detectable, although multiple activations resulted in 4-day memory (*Figure 2C*). In contrast, PPL1-$\gamma$1pedc (MB320C) activation resulted in robust immediate memory acquisition after a single round of training, but its activation failed to induce 4-day memory even after extensive spaced training (*Figure 2C*). These results imply the stability of memory is an intrinsic property of the MB compartment, rather than a consequence of the training protocol. In this view, repetitive training with naturalistic stimuli that activate many DAN cell types would recruit additional compartments with slower acquisition rates and the behaviorally assayed retention of memory would reflect the combined memories formed in different compartments (*Figure 2—figure supplement 5*). It remains an open question whether short-term memories are converted into long-term memories as biochemical changes in the same synapses or whether these memories are formed separately and in parallel. For olfactory learning in Drosophila, our data are consistent with a model in which memory formation and consolidation can occur independently and in parallel in individual MB compartments; this view does not exclude the possibility that network activity facilitates memory consolidation.

## Comparing rules for updating memory

We found that the memories induced in different compartments have different stabilities, displaying different dynamics of spontaneous memory decay over a 1-day period (*Figure 3A–B*). Memories in each compartment also differed in the extent to which they were reduced by a second presentation of the trained odor without reinforcement (*Figure 3C*).

Likewise, DAN activation without odor presentation significantly reduced immediate memory (*Figure 3D*) for four of the five sets of DANs tested. These two effects might be mechanistically linked as odor presentation alone can result in activation of a subset of dopaminergic neurons (*Riemensperger et al., 2005*; *Mao and Davis, 2009*).

In both the case of presentation of odor without dopamine and of dopamine without odor, the association the fly had previously learned is not confirmed. It would make sense for a memory to be diminished when the contingency upon which it is based is found to be unreliable. Consistent with this idea, repetitive spaced training with these same DANs can induce 1-day memory that is resistant to DAN activation (*Figure 3—figure supplement 1*). The differences we observed between compartments suggest that they weigh the importance of the reliability of the correlation between CS and US differently.

The $\alpha$1 compartment differed from the other compartments we tested in that it was resistant to memory reduction by DAN activation (*Figure 3D*). This compartment plays a key role in long-term appetitive memory of nutritious foods (*Yamagata et al., 2015*; *Huetteroth et al., 2015*) and has an unusual circuit structure: its MBON (MBON-$\alpha$1) appears to form synapses on the dendrites of the DAN that innervates the $\alpha$1 compartment (PAM-$\alpha$1) forming a recurrent circuit necessary for long-term memory formation (*Ichinose et al., 2015*). The $\alpha$1 compartment also showed the least ability to replace an older association with a new one (*Figure 3E*). This observation suggests that the initial memory may not be affected by the second training, resulting in co-existing appetitive memories for both odors. Indeed, flies were able to retain associations between each of two odors and PAM-$\alpha$1 (MB043C) activation, while only the most recently learned association was remembered with PPL1-$\gamma$

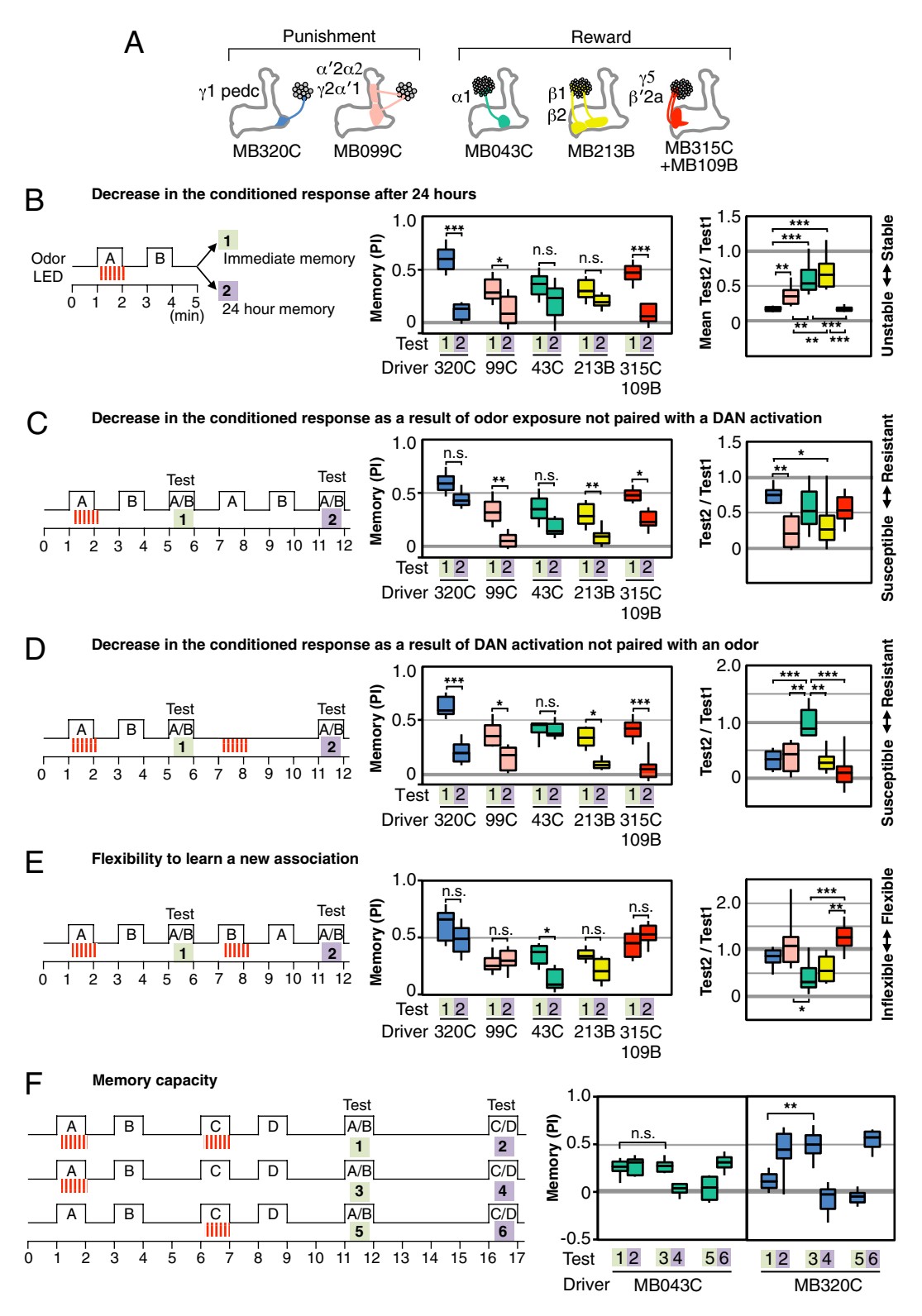

**Figure 3.** Rules for updating memory. (**A**) Diagram of the DANs contained in split-GAL4 driver lines, which have been color-coded to facilitate comparison with the plots shown in the subsequent panels. In all experiments, the drivers were crossed with 20xUAS-CsChrimson-mVenus in attP18. (**B**) Memory decay after 1 d. Left: Flies were trained with a 60-s odor delivery during which thirty 1-s pulses of red light were given, and then tested either immediately (Test 1) or after 1 d (Test 2). Center: To facilitate comparison of PI magnitudes, the sign of the PI in this and subsequent panels was

*Figure 3 continued on next page*

*Figure 3 continued*

reversed for DANs that induced aversive memory (MB320C, MB099C and MB630B). The bottom and top of each box represents the first and third quartile, and the horizontal line dividing the box is the median. The whiskers represent the 10th and 90th percentiles. N = 8–18. Right: Comparison of memory retention times induced by activation of different DANs. Ratios of the mean PI measured at 1d and individual PIs immediately after training are shown for each driver. Asterisk indicates significance of depicted pairs after comparing all pairs. (C) Decrease in the conditioned response by unpaired odor exposure. After the first training and test (Test 1), flies were exposed to both odors without optogenetic DAN activation and then retested (Test 2). In panels C–E, Test 1 and Test 2 were performed on the same group of flies and thus the ratios of individual data points are plotted in the rightmost graphs. N = 10–18. (D) Decrease in the conditioned response by unpaired DAN activation. After first training and test (Test 1), flies were exposed to light to activate DANs (thirty 1-s pulses) without odor delivery, and then retested (Test 2). N = 10–17. (E) Flexibility to learn a new association. After first training and test (Test 1), flies were trained with the opposite pairing of odor and DAN activation and tested for their ability to learn the new pairing (Test 2). N = 10–16. For most DANs, the ability to learn the second association was not significantly impaired by the first training. The exception was MB043C, which expresses in the DAN innervating the α1 compartment. (F) Memory capacity. Flies were trained and tested with two pairs of odors. Odor A and B were pentyl acetate and 3-octanol, respectively. Odor C and D were ethyl lactate and 4-methylcyclohexanol, respectively. N = 8–12. For MB320C (compartments γ1 and pedc) only the most recently trained odor was retained, an effective memory capacity of one. For MB043C (α1 compartment), flies were able to remember both comparisons, demonstrating a memory capacity of at least two. Asterisk indicates significance: n.s. not significant, *p<0.05; **p<0.01; ***p<0.001.

The following figure supplement is available for figure 3:

**Figure supplement 1.** One-day memory is resistant to unpaired DAN activation.

1pedc (MB320C) activation (3F). The higher memory capacity of the α1 compartment is not due to generalization, since training with one odor pair did not affect the innate odor preference observed with a different, untrained odor pair. Thus two distinct strategies for updating memories appear to be used in different MB compartments: (1) writing a new memory, while diminishing the old memory; or (2) writing a new memory, while retaining the old memory.

## Processing of conflicting memories

Our results suggest that memory formation in each compartment is largely parallel and independent, with compartmental specific rules for updating memories. Such a model of independent memory storage should allow appetitive and aversive memories to be simultaneously formed for the same odor in different compartments. We tested this idea by simultaneously activating DANs to α1 and γ1pedc while exposing flies to an odor (*Figure 4A*). When flies were tested immediately after training, the odor was strongly aversive, but the same odor became appetitive after 1 day. These results are most easily explained by simultaneous formation of an aversive memory in γ1pedc and an appetitive memory in α1, with rapid decay of the memory in γ1pedc and slow decay in α1 resulting in a shift in valence of the conditioned response over time. However, the fact that we observed strongly aversive immediate memory, rather than an intermediate response, suggests that the MB network non-linearly integrates these conflicting signals. The known feedforward connection between γ1pedc and α1 provides a possible circuit mechanism (*Figure 5B*; *Aso et al. 2014a*). Recent studies (*Kaun et al., 2011*; *Das et al., 2014*; *Aso et al., 2014b*) provide further examples most easily explained by parallel induction of conflicting memories of different decay rates. We also found that wild type flies are capable of efficiently switching odor preference when they had conflicting sequential experiences of sugar reward followed by shock punishment with the same odor (*Figure 4B*).

## Concluding remarks

Our results demonstrate that different MB compartments use distinct rules for writing and updating memories of odors (*Figure 5A*). By analyzing individual memory components–or engrams–induced by local dopamine release, we found that the interpretation of a common odor representation carried by sparse KC activity to multiple compartments could be modified differently in each of those compartments. We do not know the mechanisms that generate these distinct learning rules. They could arise from differences in the dopamine release properties of different DAN cell types or from local differences in the biochemical response to dopamine signaling in each MB compartment. For example, KCs express four distinct dopamine receptors (*Crocker et al., 2016*), which might be deployed differently in each compartment. Or they could originate from circuit properties: we know

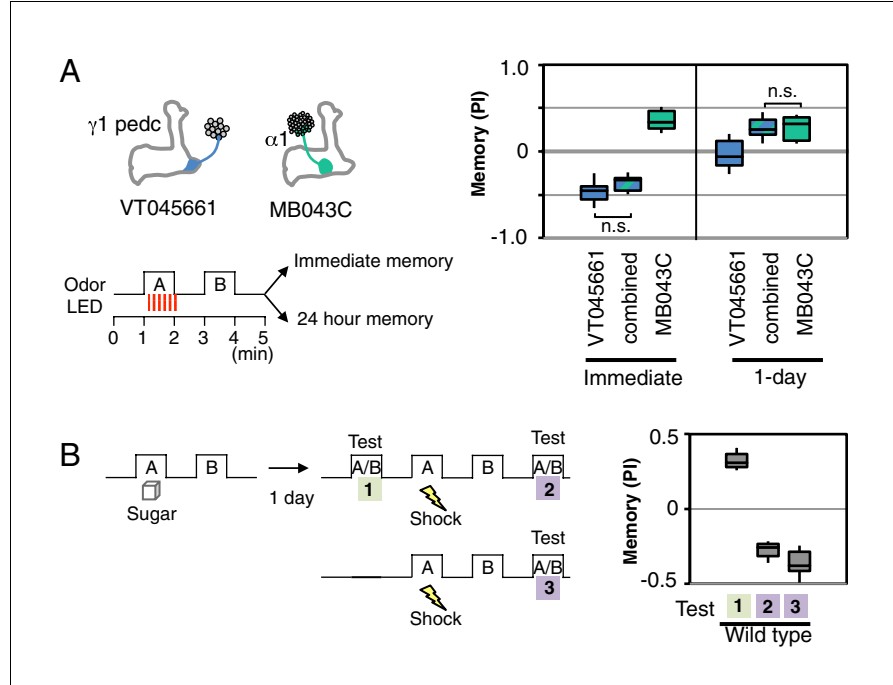

**Figure 4.** Processing of conflicting memories. (**A**) Flies expressing CsChrimson in PPL1-γ1pedc (VT045661-LexA in *JK22C* x 13xLexAop2-CsChrimson-tdTomato in *attP18*), or in PAM-α1 (MB043C x 20xUAS-CsChrimson-mVenus in attP18), or in both PPL1-γ1pedc and PAM-α1 (by combining all four transgenes) were starved for 48 hr and then trained with a 60-s odor delivery during which thirty 1-s pulses of red light were given, and then tested either immediately or after 1 d. N = 8–12. (**B**) Wild type flies were starved for 48 hr and then trained using 2-min exposures to odor A with sucrose and then to odor B without sucrose and tested for odor preference after 1d (Test 1). Following the first memory test, flies were trained with a 1-min exposure of odor A (the odor previously paired with sugar) and electric shock (twelve 1.25 s pulses of 60V) in the olfactory arena and then with odor B without shock. Odor preference was measured immediately after the second training (Test 2). For comparison, wild type flies starved for same period were conditioned with electric shock and tested immediately (Test 3). N = 8.

from anatomical (*Tanaka et al., 2008*; *Aso et al., 2014a*), behavioral (*Ichinose et al., 2015*) and functional imaging (*Boto et al., 2014*; *Cohn et al., 2015*; *Owald et al., 2015*) studies that MB compartments can communicate through connections between their extrinsic neurons, the DANs and MBONs, as well as by a layered network within the MB (*Figure 5B*). In the mammalian brain, associative memories are also stored as distributed and parallel changes with partially overlapping functions (*Herry and Johansen, 2014*; *Hikosaka et al., 2014*; *Tonegawa et al., 2015*); for example, different populations of dopaminergic neurons develop representations of a visual objects' value with distinct learning rules (*Kim et al., 2015*). We expect many of the underlying strategies and mechanisms may be shared between flies and other species. Our work provides a foundation for experiments aimed at understanding the molecular and circuit mechanisms by which distributed memory components are written with distinct rules and later integrated to guide memory-based behaviors.

## Materials and methods

### Fly strains

Crosses of split-GAL4 lines for DANs (*Aso et al., 2014a*) and 20xUAS-CsChrimson-mVenus in attP18 (*Klapoetke et al., 2014*) were kept on standard cornmeal food supplemented with retinal (0.2 mM all-trans-retinal prior to eclosion and then 0.4 mM) at 22℃ at 60% relative humidity in the dark. Female flies were sorted on cold plates at least 1 d prior to the experiments and 4–10 d old flies were used for experiments. The new split-GAL4 driver, MB630B was designed based on confocal

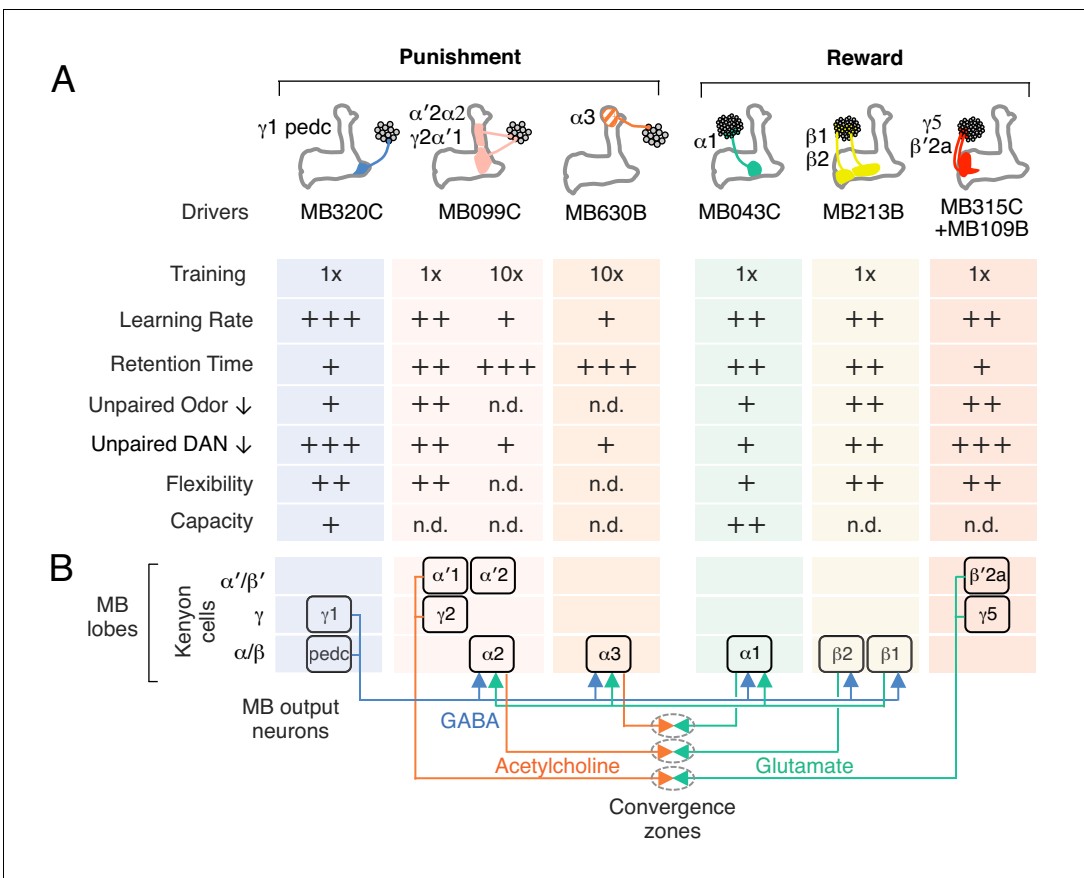

**Figure 5.** Summary of distinct rules for learning and updating memory. (**A**) Summary table of distinct learning rules induced by different DAN cell types. Criteria were qualitatively judged to be +, ++, +++ or n.d. (not determined) based on the data presented in *Figures 2,3*. (**B**) Diagram summarizing the feed forward network within the MB lobes (compartments shown as boxes) and the convergence of MBONs in common target zones in other brain areas (shown as ovals). These circuit motifs might provide a path through which memories distributed in different MB compartments might be integrated (*Aso et al., 2014a*). Not shown in this diagram are cases where MBONs target the dendrites of DANs, a circuit motif that is known to occur (*Aso et al., 2014a*; *Ichinose et al., 2015*) and could also promote communication between compartments.

image databases (http://flweb.janelia.org) (*Jenett et al., 2012*), BrainBase (http://brainbase.imp.ac.at), and constructed by inserting VT026773-p65ADZp in attP40 and R72B05-ZpGAL4DBD in attP2 as described previously (*Pfeiffer et al., 2010*). VT045661-LexA was constructed as described previously (*Pfeiffer et al., 2010*) and injected into JK22C (*Knapp et al., 2015*). The confocal images of expression patterns are available online (http://www.janelia.org/split-gal4). For driving CsChrimson by both MB109B and MB315C, 20xUAS-CsChrimson-mVenus in attP18 was first combined with MB315C, and then crossed with MB109B.

## Optogenetic olfactory arena

The olfactory arena for optogenetics experiments was designed based on the four-field olfactometer (*Pettersson, 1970*; *Vet et al., 1983*) and was briefly described in previous reports (*Aso et al., 2014b*; *Hige et al., 2015*). The overview of the assay is described in *Figure 1—figure supplement 1* and a detailed description of the apparatus is provided in *Supplementary file 1*; the Janelia Tech-Transfer office (techtransfer@janelia.hhmi.org) will provide complete construction documentation and CAD files upon request.

## Behavioral assay

Groups of approximately 20 females of 4–10 d post-eclosion were trained and tested at 25°C at 50% relative humidity in a dark chamber. The flow rate of input air from each of the four arms was maintained at 100 mL/min throughout the experiments by mass-flow controllers, and air was extracted from the central hole at 400 mL/min. Odors were delivered to the arena by switching the direction of airflow to the tubes containing diluted odors using solenoid valves. The odors were diluted in paraffin oil (Sigma–Aldrich): 3-octanol (OCT; 1:1000; Merck) and 4-methylcyclohexanol (MCH; 1:750; Sigma–Aldrich), Pentyl acetate (PA: 1:5000; Sigma–Aldrich) and ethyl lactate (EL: 1:5000; Sigma–Aldrich). Shock and sugar conditioning was performed as previously described by using tubes with sucrose absorbed Whatman 3 MM paper or copper grids (*Figure 2—figure supplement 3*) (*Aso et al., 2012*; *Liu et al 2012*). For the experiments in *Figure 4B*, a sheet of copper grid was placed at the bottom of arena. For appetitive memory assays, flies were starved for 24–48 hr on 1% agar. Videography was performed at 30 frames per second and analyzed using Fiji (*Schindelin et al., 2012*). Statistical comparisons were performed (Prism; Graphpad Inc, La Jolla, CA 92037) using the Kruskal Wallis test followed by Dunn's post-test for multiple comparison, except those in *Figure 1F*, *Figure 2C* and *Figure 2—figure supplement 4* which used Wilcoxon signed-rank test with Bonferroni correction to compare from zero.

## Acknowledgements

We thank Igor Negrashov, Steven Sawtelle, Peter Polidoro, William Rowell, Jinyang Liu, Alice Robie, Kristin Branson, Chuntao Dan, Roman Huber and Hiromu Tanimoto for help in establishing the olfactory arena. Brandi Sharp, James McMahon, Lori Laughrey, Teri Ngo and the Janelia Fly Facility helped in fly husbandry. Rebecca Vorimo, Allison Sowell and the FlyLight Project Team performed brain dissections and histological preparations. VT026773-p65ADZp was a gift from Barry J Dickson. Heather Dionne made new molecular constructs. We thank Gowan Tervo, Joshua Dudman, Ulrike Heberlein, TJ Florence, Yichun Shuai, Kit Longden, Adam Hantman, Daisuke Hattori, Larry Abbott, Richard Axel, Chuntao Dan, Krystyna Keleman, Toshihide Hige, Glenn Turner, Toshiharu Ichinose, Nobuhiro Yamagata and Hiromu Tanimoto for stimulating discussions and for comments on earlier drafts of the manuscript.

## Additional information

### Funding

| Funder | Author |
| --- | --- |
| Howard Hughes Medical Institute | Yoshinori Aso<br>Gerald M Rubin |

The funders had no role in study design, data collection and interpretation, or the decision to submit the work for publication.

### Author contributions

YA, Conceived and designed the study, Acquired and analyzed the data, Analysis and interpretation of data, Drafting or revising the article, Wrote the article; GMR, Conceived and designed the study, Analysis and interpretation of data, Drafting or revising the article, Wrote the article

### Author ORCIDs

Yoshinori Aso, http://orcid.org/0000-0002-2939-1688
Gerald M Rubin, http://orcid.org/0000-0001-8762-8703

## Additional files

### Supplementary files
• Supplementary file 1. Design and parts list of the olfactory behavioral apparatus. The first page shows a side view of the apparatus with a parts list. On the second page, a 3D model of the apparatus is shown which can be rotated and the visualization of each part can be individually turned on or off.

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
