## [Decision Letter]

Thank you for submitting your article "Dopaminergic neurons write and update memories with cell-type-specific rules" for consideration by *eLife*. Your article has been reviewed by three peer reviewers, and the evaluation has been overseen by a Reviewing Editor, Liqun Luo, and K VijayRaghavan as the Senior Editor. The following individuals involved in review of your submission have agreed to reveal their identity: Josh Dubnau (Reviewer #1); Yi Zhong (Reviewer #2); Leslie C Griffith (Reviewer #3).

The reviewers have discussed the reviews with one another and the Reviewing Editor has drafted this decision to help you prepare a revised submission.

All three reviewers, as well as the Reviewing Editor, are highly enthusiastic about the significance, and most of the critiques can be addressed by textual changes. However, the first critique of Reviewer 1 regarding back-crossing represents substantial amount of work if the current data were collected from flies that had not been back-crossed to a common genetic background. We hope that these flies were indeed back-crossed and it was an omission in the methods. If they had not been, after much discussion among the Reviewers and Reviewing Editor, we think that it is necessary to perform additional controls to exclude the possibility that the differences reported in Figure 2,Figure 3 are due to different backgrounds of flies harboring individual transgenes.

[We note that in the Aso et al., 2014 paper, you used many conditional manipulations (e.g. shibire). So even though each Gal4 line or split may be on a unique background, you had a within genotype temperature shift manipulation so we could see the acute impact of that cell type. Here this is not the case, and there are direct comparisons between lines and conclusions drawn in a comparative sense between the cell types.]

Specifically, we suggest that you perform standard electric shock/sugar training paradigms with each of the key strains (Chrimson included), and examine short term and long term memory. If each of these strains you use has comparable conditioned responses to electric shock and sugar, we can be more certain that the differences with light-induced training are due to cell-type specific Chr expression, not something in the strain background.

Reviewer #1:

The MB are a highly studied neural center of associative olfactory memory and learning in flies. A large body of prior work establishes that odor driven CS inputs are represented as sparse patterns of activity within the large ensemble of MB Kenyon cslls. US driven dopaminergic neuron inputs are thought to converge onto MB Kenyon cells in the MB lobes, which is the site of Kenyon cell synapses with output neurons. Recent work has established that the different sub-types of dopaminergic neurons (DANs) and output neurons (MBONs) are each restricted within zones of the MB lobes. The anatomical organization of the MB Kenyon cell sub-types also implies that the DAN/MBON zones are restricted to sub-types of Kenyon cells. Layered onto this anatomical model is a large body of work that demonstrates that memory is processed through a series of genetically dissectible phases, that rely on different sub-sets of the MB Kenyon cells. This manuscript investigates very interesting questions about how the anatomical organization of the DANs and MBONs into sub-zones provide functional modules upon which each of the memory phases after appetitive vs aversive memories are built. The questions being asked are important and the world view that the authors propose is beautiful. So on a deep level, I am highly enthusiastic. Unfortunately, however, I have several foundational critiques that undermine the major conclusions of this study. This is really unfortunate, because I think that if properly done, this could be a really important piece of work. Below I outline in roughly descending order of importance, the issues that I have with this study in its current form.

1) If my read of the Methods section is accurate, the authors have made no attempt to control the genetic backgrounds in any of their experiments. This is a fundamental and in my view fatal flaw. There is a long and rich history of this, and it is well established that for all quantitative behavioral traits, genetic background needs to be controlled between all groups for which differences in phenotype are used to draw conclusions. In this case, the authors have used Gal4, and Gal4 reagents that were generated on an outbred background. So every single line has a founder effect, which means that the genetic background of every single strain used is completely unique. As a result, all bets are off in terms of how we interpret differences in magnitude of performance, kinetics of memory decay, and even olfactory acuity and locomotion behaviors that underlie the memory performance. Any and all of the 'rules' that are discovered here that govern how activation of a given DAN might contribute to short term or long-term memory, or to its extinction, etc., could just as easily be due to differences in the kinetics and processing of memory between genetic backgrounds. If I had to guess from my own personal experience, I would wager that a decent fraction of observed quantitative differences between different DANs will fail to reproduce when the genetic backgrounds are equilibrated. We simply cannot fill the literature with this sort of doubt.

2) The authors state up front that strong optogenetic activation of a single class of neuron may not reflect real physiological US inputs. Although some reviewers may have an issue with that, I actually have no conceptual problem with it: I buy the argument that this is a good framework by which to pick apart the rules that govern plasticity at each DAN:KC:MBON zone. However, it is essential to do this sort of analysis in parallel with 'real' learning in which a real US is used (sugar reward, electric shock or some other aversive stimulus). And then within the context of a real US, it is important to query the contribution of that DAN to performance. Otherwise, the insight that we gain from activating a DAN in place of the US is severely reduced.

Critiques that need to be addressed with text revisions:

3) The major dopamine receptor that is established to mediate the US inputs is the type 1 receptor, DopR. Mutations in DopR can be rescued by expression only in MB (Kim et al) and in the case of aversive memory, expression of this receptor only in MB γ neurons is sufficient to fully restore all stages of memory. Given this, how do the authors interpret the fact that activation of DANs that are restricted to zones outside γ lobes are able to induce aversive memories? Is there a different Dopamine receptor at play? Or should we conclude that DANs also co-release other transmitters that are not acting through a dopamine receptor? I am not arguing that the authors need to resolve this issue experimentally, but they cannot ignore this.

4) Long term memory in this field has an operational meaning that is mis-applied here. When this field uses this term, we mean memory that requires new protein synthesis. In general, it has been required for any new behavioral assay that one use inhibitors of protein synthesis and/or establish a requirement for the CREB gene and/or show a difference between massed versus spaced training. Here, the authors use a spaced training protocol, and then refer to a long lived trace as long-term memory. But we really don't know whether this is the same consolidated form of memory that others have discussed. Moreover, the text confounds the impact of multiple training sessions on acquisition versus consolidation.

5) The accepted historical nomenclature for Pavlovian literature is flaunted badly. Extinction has a specific meaning in the literature, and it does not involve presentation of the US absent the CS. So activation of DANs after training should not be called extinction, and it shouldn't be said to extinguish memory. And even in the cases where CS+ is presented post training, it is not clear that the decrement in performance is due to extinction. If I read the methods correctly, this is a degradation of avoidance under continuous odor exposure. So this easily could be due to (e.g.) sensory adaptation. Extinction has specific definitions, and there are behavioral tests for whether extinction has taken place. Similarly, the term 'relieve learning' is not really correct. I realize that this term has been used by Tanimoto et al. But the larger literature on Pavlovian learning has already established terms: conditioned excitation and conditioned inhibition. Relief learning is a rediscovery of CI and CE.

6) The discussion of parallel vs. sequential memory formation is superficial. Isabel et al. and Blum et al. do not impact this issue in the way the authors have implied, and I don't think the findings in this paper bear on this discussion in a compelling way.

In sum, I very much like the outlook and the over-all approach. But this manuscript does not provide compelling data to support the main conclusions. I realize that the bar I am setting would require (e.g.) back-crossing all lines to the same strain and then repeating most or all of the experiments. But this bar has been set, appropriately, by a consensus in this field that conclusions that are drawn from behavioral comparisons with uncontrolled genetic backgrounds do not withstand the test of time. We all live by this standard.

Reviewer #2:

In this manuscript, the authors systematically analyzed the impacts of artificial activation of several DAN subsets on olfactory memory formation, retention, extinction, extinguishment and reverse. The majority of their discoveries are new while a few observations are confirmative of previously published reports. All observations presented in this manuscript together help to clarify the roles of DANs in different memory processes and memory components and build a systematic understanding of involvement of the dopaminergic system in memory. This manuscript is suitable for publication in *eLife* if the following concerns are addressed:

1) The statement concerning "extinguish" needs to be clarified. To show whether the first odor memory is extinguished, the authors may want to perform a third odor test (test the preference between the first odor and a third odor).

2) The manuscript included 1d and 4d odor memories. It would be of interests to determine whether these memories are long-term one or not through feeding of flies with CXM.

3) The grammatical mistakes should be corrected, such as "each innervate distinct MB compartments" and "a series of experiences is integrated to make a probabilistic prediction".

4) Some technical terms, such as "*Drosophila*" and Gal4 names, should be written in the right format.

Reviewer #3:

This paper is an interesting and important contribution to our understanding of how dopamine can influence behavior. The last few years have seen a number of high profile papers that suggest that particular subsets of DAergic neurons can have specialized roles in either temporal or functional domains. This paper steps back and takes a very broad view of this issue looking at most of the DAergic neurons and testing their roles in multiple temporal scales in the encoding and extinguishing of appetitive and aversive memories.

The results are important for the field in two ways. First, they demonstrate that essentially all DAergic neurons have some specialist qualities. The previously described "forgetting neurons" and "STM" and "LTM" neurons are not outliers, but rather reflect a basic feature of the system. Second, they provide very strong evidence to buttress the idea that there are multiple parallel streams of memory formation. Short-term memories are not simply converted to long-term memories, but rather they are independently encoded. This paper is well-suited for this *eLife* format since these data are directly tied to the previous Aso papers and the tools they generated.

My only comments regard putting these data in context of older findings:

1) The figure legends have way too much text. Descriptions of data not presented in a figure should be in the Results section.

2) A subset of PPL1 neurons are activated by odors (Mao & Davis 2009). How does this affect the interpretation of Figure 3 (odor extinction) and Figure 3 (US extinction)?

3) Yarali & Gerber (2010) found that TH-gal4 neurons are not necessary for relief learning. Does this study conflict with these prior findings (i.e., in finding that gamm1-pedc is sufficient for relief learning), or do you propose that there are both DAN and non-DAN mediated relief learning pathways?

---

## [Author Response]

*All three reviewers, as well as the Reviewing Editor, are highly enthusiastic about the significance, and most of the critiques can be addressed by textual changes. However, the first critique of Reviewer 1 regarding back-crossing represents substantial amount of work if the current data were collected from flies that had not been back-crossed to a common genetic background. We hope that these flies were indeed back-crossed and it was an omission in the methods. If they had not been, after much discussion among the Reviewers and Reviewing Editor, we think that it is necessary to perform additional controls to exclude the possibility that the differences reported in Figure 2,Figure 3 are due to different backgrounds of flies harboring individual transgenes.*

*[We note that in the Aso et al., 2014 paper, you used many conditional manipulations (e.g. shibire). So even though each Gal4 line or split may be on a unique background, you had a within genotype temperature shift manipulation so we could see the acute impact of that cell type. Here this is not the case, and there are direct comparisons between lines and conclusions drawn in a comparative sense between the cell types.]*

Specifically, we suggest that you perform standard electric shock/sugar training paradigms with each of the key strains (Chrimson included), and examine short term and long term memory. If each of these strains you use has comparable conditioned responses to electric shock and sugar, we can be more certain that the differences with light-induced training are due to cell-type specific Chr expression, not something in the strain background.

We performed the suggested experiments. These data, which confirm that the strains have comparable conditioned responses, have been added as Figure 2—figure supplement 3.

*Reviewer #1:*

*The MB are a highly studied neural center of associative olfactory memory and learning in flies. A large body of prior work establishes that odor driven CS inputs are represented as sparse patterns of activity within the large ensemble of MB Kenyon cslls. US driven dopaminergic neuron inputs are thought to converge onto MB Kenyon cells in the MB lobes, which is the site of Kenyon cell synapses with output neurons. Recent work has established that the different sub-types of dopaminergic neurons (DANs) and output neurons (MBONs) are each restricted within zones of the MB lobes. The anatomical organization of the MB Kenyon cell sub-types also implies that the DAN/MBON zones are restricted to sub-types of Kenyon cells. Layered onto this anatomical model is a large body of work that demonstrates that memory is processed through a series of genetically dissectible phases, that rely on different sub-sets of the MB Kenyon cells. This manuscript investigates very interesting questions about how the anatomical organization of the DANs and MBONs into sub-zones provide functional modules upon which each of the memory phases after appetitive vs aversive memories are built. The questions being asked are important and the world view that the authors propose is beautiful. So on a deep level, I am highly enthusiastic. Unfortunately, however, I have several foundational critiques that undermine the major conclusions of this study. This is really unfortunate, because I think that if properly done, this could be a really important piece of work. Below I outline in roughly descending order of importance, the issues that I have with this study in its current form.*

*1) If my read of the Methods section is accurate, the authors have made no attempt to control the genetic backgrounds in any of their experiments. This is a fundamental and in my view fatal flaw. There is a long and rich history of this, and it is well established that for all quantitative behavioral traits, genetic background needs to be controlled between all groups for which differences in phenotype are used to draw conclusions. In this case, the authors have used Gal4, and Gal4 reagents that were generated on an outbred background. So every single line has a founder effect, which means that the genetic background of every single strain used is completely unique. As a result, all bets are off in terms of how we interpret differences in magnitude of performance, kinetics of memory decay, and even olfactory acuity and locomotion behaviors that underlie the memory performance. Any and all of the 'rules' that are discovered here that govern how activation of a given DAN might contribute to short term or long-term memory, or to its extinction, etc., could just as easily be due to differences in the kinetics and processing of memory between genetic backgrounds. If I had to guess from my own personal experience, I would wager that a decent fraction of observed quantitative differences between different DANs will fail to reproduce when the genetic backgrounds are equilibrated. We simply cannot fill the literature with this sort of doubt.*

We agree that difference in genetic background can influence behavioral phenotypes, especially for long-term memory. However, we note that the split-GAL4 stocks that we used are much more similar in background than the stocks that the reviewer correctly notes have proved problematic in the past. Those stocks were often genetic mutations or enhancer trap GAL4 lines in widely different genetic backgrounds. While the split-GAL4 lines we used were not back-crossed, these lines are in similar genetic backgrounds: the AD and DBD DNA constructs used in the split-GAL4 lines were injected into the same set of recipient stocks and inserted into the same landing sites, and the X chromosome in all split-GAL4 lines was exchanged with that of a common stock. Also, unlike in many previous experiments that studied the behavioral effects of mutations, we do not test these split-GAL4 lines as homozygotes, but as a population of the heterozygous progeny of a cross between a specific split-GAL4 line and a common 20xUAS-CsChrimson-mVenus stock, thus masking the effect of any recessive genetic differences between the split-GAL4 stocks.

2) The authors state up front that strong optogenetic activation of a single class of neuron may not reflect real physiological US inputs. Although some reviewers may have an issue with that, I actually have no conceptual problem with it: I buy the argument that this is a good framework by which to pick apart the rules that govern plasticity at each DAN:KC:MBON zone. However, it is essential to do this sort of analysis in parallel with 'real' learning in which a real US is used (sugar reward, electric shock or some other aversive stimulus). And then within the context of a real US, it is important to query the contribution of that DAN to performance. Otherwise, the insight that we gain from activating a DAN in place of the US is severely reduced.

We agree that studying the effects of blocking specific DANs after presentation of a “real” US (shock or sugar) will be needed to fully understand associative learning. However, interpreting the results of such experiments is complicated by the fact that shock or sugar activate multiple DANs (Kirkhart, C. & Scott, K. 2015; Mao, Z. & Davis, R. L. 2009; Liu, C. et al. 2012) and almost certainly, given previous studies (Aso et al., 2012) and the results we present in this paper, induce partially redundant memories in multiple compartments in parallel. Thus making conclusions about the roles of individual DANs by studying the behavioral effects of their inactivation is not straightforward and would, at a minimum, require construction of strains that would allow different, specific combinations of DANs to be inactivated together. These experiments are technically challenging and beyond the scope of the current study. Indeed, avoiding the complications of redundancy was a major motivation of our current study’s experimental design: inducing memory in defined compartment(s) by optogenetic stimulation and then assaying memory dynamics using behavioral assays.

*Critiques that need to be addressed with text revisions:*

*3) The major dopamine receptor that is established to mediate the US inputs is the type 1 receptor, DopR. Mutations in DopR can be rescued by expression only in MB (Kim et al) and in the case of aversive memory, expression of this receptor only in MB γ neurons is sufficient to fully restore all stages of memory. Given this, how do the authors interpret the fact that activation of DANs that are restricted to zones outside γ lobes are able to induce aversive memories? Is there a different Dopamine receptor at play? Or should we conclude that DANs also co-release other transmitters that are not acting through a dopamine receptor? I am not arguing that the authors need to resolve this issue experimentally, but they cannot ignore this.*

In Figure 2, we showed that PPL1-α3 can induce 1-day and 4-day memory after 10x spaced training. These data fit well with previous reports that *rutabaga* rescue in α/β KCs can restore LTM (Blum 2010), but appear at first glance to be inconsistent with the results reported in Qin et al. 2012 that showed full restoration of both immediate and 1-day memory in DopR1 mutants by driving DopR1 in γ KCs but not in α/β KCs. There are many possible explanations for such lack of agreement, including the use of other dopamine receptors in α/β Kenyon cells. Indeed, α/β and γ Kenyon cells have been reported to express high levels of four different dopamine receptors (Crocker et al., 2016). We added a statement about the presence of multiple dopamine receptors to the text.

4) Long term memory in this field has an operational meaning that is mis-applied here. When this field uses this term, we mean memory that requires new protein synthesis. In general, it has been required for any new behavioral assay that one use inhibitors of protein synthesis and/or establish a requirement for the CREB gene and/or show a difference between massed versus spaced training. Here, the authors use a spaced training protocol, and then refer to a long lived trace as long-term memory. But we really don't know whether this is the same consolidated form of memory that others have discussed. Moreover, the text confounds the impact of multiple training sessions on acquisition versus consolidation.

We have modified the text to use the simple operational description of “4-day” memory throughout, rather than “long-term” memory, which we agree implies specific attributes that were not assessed by our experiments.

*5) The accepted historical nomenclature for Pavlovian literature is flaunted badly. Extinction has a specific meaning in the literature, and it does not involve presentation of the US absent the CS. So activation of DANs after training should not be called extinction, and it shouldn't be said to extinguish memory. And even in the cases where CS+ is presented post training, it is not clear that the decrement in performance is due to extinction. If I read the methods correctly, this is a degradation of avoidance under continuous odor exposure. So this easily could be due to (e.g.) sensory adaptation. Extinction has specific definitions, and there are behavioral tests for whether extinction has taken place. Similarly, the term 'relieve learning' is not really correct. I realize that this term has been used by Tanimoto et al. But the larger literature on Pavlovian learning has already established terms: conditioned excitation and conditioned inhibition. Relief learning is a rediscovery of CI and CE.*

We removed the terms “extinction” and “relief learning” from the text. Instead we simply describe our experiments in operational terms.

*6) The discussion of parallel vs. sequential memory formation is superficial. Isabel et al. and Blum et al. do not impact this issue in the way the authors have implied, and I don't think the findings in this paper bear on this discussion in a compelling way.*

We have modified this section of the text and removed these two citations. We agree that our data to not resolve the issue of parallel vs. sequential memory formation and now simply say that our results “provide support for” parallel learning.

We also added an additional experiment that we feel bears directly on the issue of parallel memory (new Figure 4).

*Reviewer #2:*

*In this manuscript, the authors systematically analyzed the impacts of artificial activation of several DAN subsets on olfactory memory formation, retention, extinction, extinguishment and reverse. The majority of their discoveries are new while a few observations are confirmative of previously published reports. All observations presented in this manuscript together help to clarify the roles of DANs in different memory processes and memory components and build a systematic understanding of involvement of the dopaminergic system in memory. This manuscript is suitable for publication in eLife if the following concerns are addressed:*

*1) The statement concerning "extinguish" needs to be clarified. To show whether the first odor memory is extinguished, the authors may want to perform a third odor test (test the preference between the first odor and a third odor).*

We removed the terms “extinction” and “relieve learning” from the text. Instead we simply describe our experiments in operational terms.

*2) The manuscript included 1d and 4d odor memories. It would be of interests to determine whether these memories are long-term one or not through feeding of flies with CXM.*

We have modified the text to use the simple operational description of “4-day” memory throughout, rather than “long-term” memory, which we agree implies specific attributes that were not assessed by our experiments.

*3) The grammatical mistakes should be corrected, such as "each innervate distinct MB compartments" and "a series of experiences is integrated to make a probabilistic prediction".*

Done. (“Series” is a singular noun.)

4) Some technical terms, such as "Drosophila" and Gal4 names, should be written in the right format.

Done.

*Reviewer #3:*

*This paper is an interesting and important contribution to our understanding of how dopamine can influence behavior. The last few years have seen a number of high profile papers that suggest that particular subsets of DAergic neurons can have specialized roles in either temporal or functional domains. This paper steps back and takes a very broad view of this issue looking at most of the DAergic neurons and testing their roles in multiple temporal scales in the encoding and extinguishing of appetitive and aversive memories.*

*The results are important for the field in two ways. First, they demonstrate that essentially all DAergic neurons have some specialist qualities. The previously described "forgetting neurons" and "STM" and "LTM" neurons are not outliers, but rather reflect a basic feature of the system. Second, they provide very strong evidence to buttress the idea that there are multiple parallel streams of memory formation. Short-term memories are not simply converted to long-term memories, but rather they are independently encoded. This paper is well-suited for this eLife format since these data are directly tied to the previous Aso papers and the tools they generated.*

*My only comments regard putting these data in context of older findings:*

1) The figure legends have way too much text. Descriptions of data not presented in a figure should be in the Results section.

We moved some text from figure legends into Results as suggested.

2) A subset of PPL1 neurons are activated by odors (Mao & Davis 2009). How does this affect the interpretation of Figure 3 (odor extinction) and Figure 3 (US extinction)?

We agree that the results of Mao & Davis (2009) showing that odor presentation alone can result in activation of a subset of dopaminergic neurons (different to those that we studied here) raises the possibility that activated dopaminergic neurons might play a role in the phenotype we report in Figure 3. We have added a comment to that effect in the text.

*3) Yarali & Gerber (2010) found that TH-gal4 neurons are not necessary for relief learning. Does this study conflict with these prior findings (i.e., in finding that gamm1-pedc is sufficient for relief learning), or do you propose that there are both DAN and non-DAN mediated relief learning pathways?*

This apparent conflict could be explained by redundancy among DANs, since neither TH-GAL4 nor DDC-GAL4 include all the DANs to MB.